# Ventilator-associated pneumonia among ICU patients in WHO Southeast Asian region: A systematic review

Sanjeev Kharel[1][⊚]*, Anil Bist[1][⊚], Shyam Kumar Mishra[2]

1 Maharajgunj Medical Campus, Tribhuvan University Institute of Medicine, Kathmandu, Nepal,
2 Department of Microbiology, Institute of Medicine, Tribhuvan University, Kathmandu, Nepal

⊚ These authors contributed equally to this work.
* kharel_sanjeev@iom.edu.np

**Data Availability Statement:** All relevant data are within the manuscript and its Supporting Information files.

**Funding:** The author(s) received no specific funding for this work.

## Abstract

Ventilator-associated pneumonia (VAP) is one of the most frequent ICU-acquired infections and a leading cause of death among patients in Intensive Care Unit (ICU). The South East Asian Region is a part of the world with limited health resources where infectious diseases are still underestimated. We aimed to review the literature in this part of the world to describe incidence, mortality and microbiological evidence of VAP and explore preventive and control strategies. We selected 24 peer-reviewed articles published from January 1, 2000 to September 1, 2020 from electronic databases and manual searching for observational studies among adult patients diagnosed with VAP expressed per thousand days admitted in ICU. The VAP rates ranged from 2.13 to 116 per thousand days, varying among different countries of this region. A significant rate of mortality was observed in 13 studies ranging from 16.2% to 74.1%. Gram negative organisms like Acinetobacter spp., Pseudomonas aeruginosa and Klebsiella pneumoniae and Gram-positive organisms like Staphylococcus aureus and Enterococcus species were frequently found. Our findings suggest an alarming situation of VAP among patients of most of the countries of this region with increasing incidence, mortality and antibiotic resistance. Thus, there is an urgent need for cost effective control and preventive measures like interventional studies and educational programs on staff training, hand hygiene, awareness on antibiotic resistance, implementation of antibiotic stewardship programs and appropriate use of ventilator bundle approach.

## Introduction

Ventilator-associated pneumonia (VAP) is defined as pneumonia or infection in lung parenchyma acquired in patients after invasive mechanical ventilation after 48–72 hours. New or progressive infiltrates, systemic infection (fever, altered white blood cell counts), changes in sputum characteristics, and the detection of a causative agent are seen in VAP patients [1]. VAP is the most common ICU acquired pneumonia among invasive mechanically ventilated patients [2]. VAP is recognized as a major issue worldwide, and common healthcare-associated infection(HAI) among developing countries associated with mortality, longer length of

**Competing interests:** The authors have declared that no competing interests exist.

stay, and associated cost burden among patients [3–5]. There is variability in the VAP rate in different studies caused by differing diagnostic criteria, ICUs type, patients' characteristics, and also varying causative microorganisms associated with patients' characteristics, length of stay, and antibiotic use in hospitals [6].

VAP risk factors include oropharyngeal and gastric colonization, thermal injuries; post-traumatic, postsurgical intervention factors such as emergency intubation, reintubation, tracheostomy, bronchoscopy and inserting nasogastric tube; patients' body positioning, level of consciousness, stress ulcer prophylaxis, and use of medications, including sedative agents, immunosuppression and antibiotics [7,8]. There is still confusion in VAP's epidemiology and diagnostic criteria, although of great advancement in microbiological tools and antimicrobials regimen for treatment. This has affected rapid diagnosis and treatment with suitable antibiotics deteriorating patient's prognosis and increase in the number of new multi-drug resistant pathogens (MDR) [2,9].

HAI are still underestimated in resource-limited countries. Although the economics of low- and middle-income countries of Asia are rapidly developing but still there is gradual advancement in the health sector causing limited access to modern health facilities to increasing population. Improvement in infection-control practices and surveillance systems can improve the safety and reduce the occurrence of adverse events among patients [10,11]. Most countries in the South-East Asian Region according to World Health Organization (WHO SEAR) still have a high burden of infectious disease even after years of development and rise in the economy [12]. Although there are little studies on countries of WHO SEAR region on this critical issue of ICU, there is a lack of rigorously carried out analytical data and reviews in this region. The main objective of this research is to estimate the incidence, mortality, and etiological agents of VAP. This reliable and updated data would be help for assessing the gravity of the situation, providing evidence for patients, clinicians and policy makers for planning infection control and other prevention strategies along with interventional educational programs.

## Materials and methods

### Search strategy

We conducted an extensive literature search of the three electronic databases, namely PubMed, Embase, and Google Scholar, to identify all the peer-reviewed research articles published within the time frame of January 1 2000 to September 1 2020. The search strategy is given in S1 File. All the databases were searched using relevant MeSH Terms and Emtree terms in PubMed and Embase, respectively. The terms "Ventilator-associated Pneumonia"," Bacterial Pneumonia", "Microbiology", "Mechanical ventilation" were searched under MeSH terms along with the names of different countries that belong to the WHO SEAR. All the references to the studies qualified for the review were also thoroughly searched for additional relevant articles. Our systematic review was not previously registered with PROSPERO or any of the international Systematic Review Registers.

### Eligibility criteria

Studies published in English (observational studies: surveillance, retrospective, and prospective studies) including information on at least prevalence or incidence or incidence rate of Ventilator associated pneumonia among adults expressed as episodes per 1000 ventilator days were considered eligible for inclusion. Besides, studies conducted in the WHO SEAR region examining mechanically ventilated adults in ICU were included.

**The following exclusion criteria were included:**

- Review articles, research protocols

- case series/case reports

- symposium/conference proceedings, commentaries/editorials/letters, views/opinions

- studies with unclear study designs and unavailable data for risk calculation.

- Full text unavailable,

- Not in English

For two or more studies, including the same set of patients, we included the study with a more significant number of patients. The PRISMA diagram detailing the identification and selection process is given in Fig 1.

## Data extraction and management

According to prespecified inclusion and exclusion criteria, two independent authors (SK and AB) screened the articles remaining after duplicates removal. Full-text articles were retrieved, and studies were shortlisted to be included in the review, which met the eligibility criteria. For full text and missing data, authors of respective studies were contacted via email and Research-Gate. The disparities and confusions among two authors were resolved by consultation of a third author (SKM). The data was extracted using the data abstraction Spreadsheet in Microsoft Excel version 2013(Microsoft Corp., Redmond, WA, USA) under the following variables: Name of the Author, Country where the study was done, Year of the Publication, Type of Study, Type of the ICU, Criteria used for diagnosis of Ventilator-Associated pneumonia (VAP), Incidence of VAP, the Mortality rate in the VAP patients and microbiological profile.

## Results

### Study selection

A total of 367 articles were obtained after a thorough search through the databases. After adjusting all the duplicate articles, a total of 303 articles remained for further processing. After the title and abstract screening, 236 of those papers were omitted because they did not follow the inclusion requirements. The full text was obtained for the remaining 67 articles. Of the 67 full-text posts, 43 were rejected because the findings of interest were not found. Finally, 24 articles were included in the review.

### Study characteristics

Altogether 24 studies (prospective, retrospective, and surveillance study) conducted in hospital settings among different adult patients presented in ICU were reviewed qualitatively. Studies included US Centers for Disease Control and Prevention (CDC) and Clinical Pulmonary Infection Score (CPIS) as diagnostic criteria for VAP expressed per thousand ventilator days. Countries of WHO SEAR (Bangladesh, India, South Korea, Nepal and Thailand) were study settings for the study. While patients of five countries of WHO SEAR (Myanmar, Maldives, Timor-Leste, Bhutan, Indonesia, and Sri Lanka) with no data on VAP were not included in the review. A detailed description of the characteristics of individual studies is provided in Table 1.

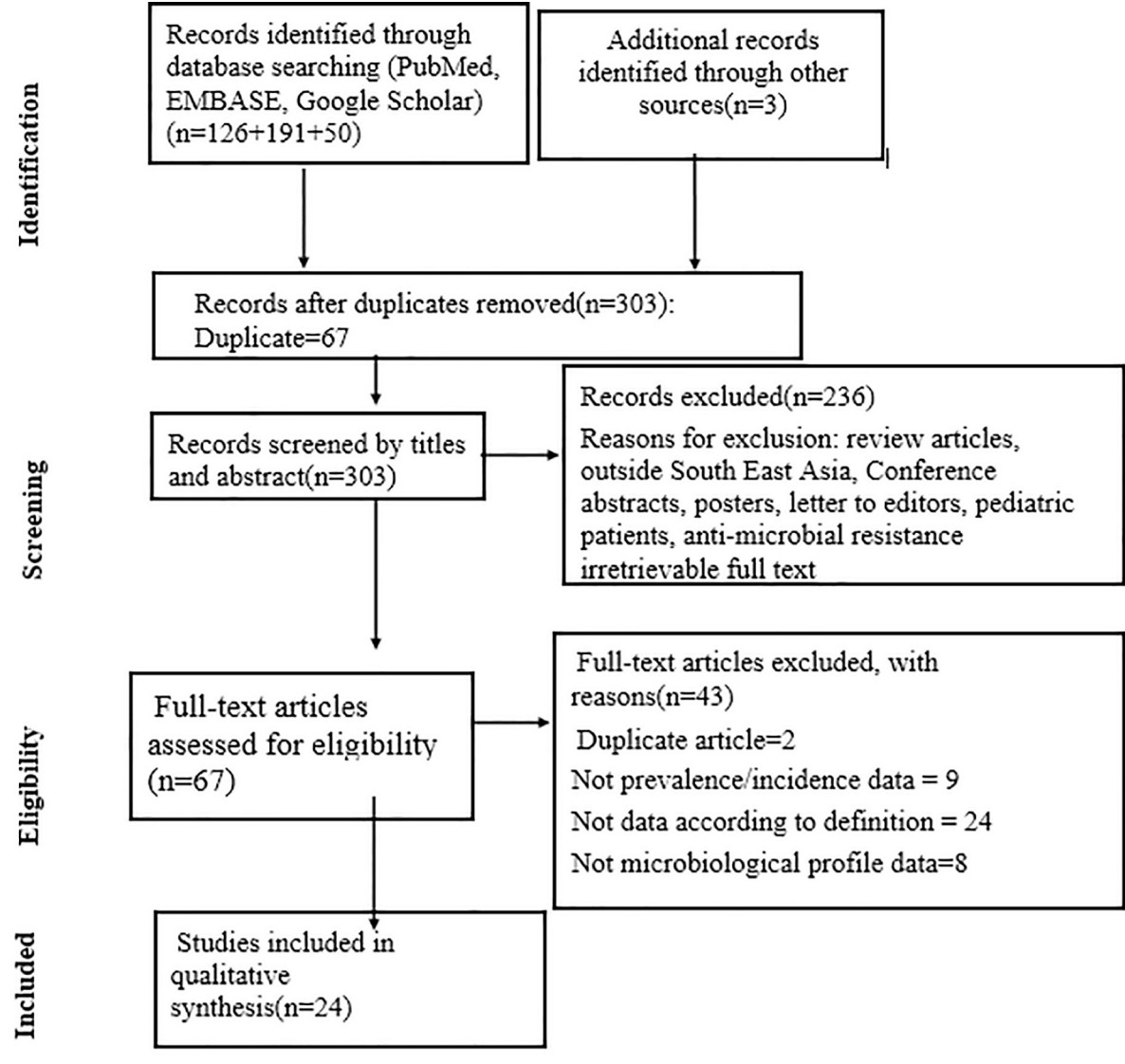

**Fig 1. The PRISMA diagram detailing selection of studies as per eligibility criteria.**

## Outcome

**Ventilator-associated pneumonia incidence rate.** The VAP incidence rate ranged from 2.13 per thousand ventilator days to 116 per thousand ventilator days differing greatly between countries. The highest VAP prevalence rate was reported from the Medical Intensive Care Unit (MICU), India, whereas the lowest was from the Palliative Care ICU setting, South Korea. The VAP rate reported from various studies are reported in Table 1.

**Mortality.** Thirteen of the 24 articles included in the review reported the mortality rate in VAP patients. The mortality rate ranged from 16.2% to 74.17%. The highest mortality rate was reported from a study of India. No mortality rates were reported from studies of Thailand and Korea. The detailed description of the mortality rate reported from studies of different countries is shown in Table 1.

**Table 1. Study characteristics of included studies.**

| Study name | Year | Study design | Country | Type of ICU | Criteria used to diagnose | VAP rate i.e. Episodes per 1000 ventilator days | Mortality (%) |
|---|---|---|---|---|---|---|---|
| Chittawatanarat et al. [13] | 2014 | Prospective study | Thailand | SICU | CPIS | 6.3±2.8 | 30.7 |
| Dasgupta et al. [14] | 2015 | Prospective study | India | MICU and SICU | CDC | 26.6 | NA |
| Datta et al. [15] | 2014 | Prospective study | India | NA | CDC | 6.04 | NA |
| Joseph et al. [16] | 2009 | Prospective study | India | MICU and CCU | CPIS | 22.94 | 16.2 |
| Khan et al. [17] | 2017 | Prospective study | India | MICU and SICU | CDC | 14.35± 8.1 | 30 |
| Khurana et al. [18] | 2017 | Prospective study | India | Neurosurgery and Polytrauma | CDC | 11.9 | 23.21 |
| Kumar et al. [19] | 2017 | Surveillance study | India | MICU and SICU | CDC | 11.8 | NA |
| Mallick et al. [20] | 2015 | Prospective cohort | Bangladesh | CCU | CDC | 35.73 | 44 |
| Maqbool et al. [21] | 2017 | Prospective study | India | MICU | CPIS | 17.09 | 50 |
| Masih et al. [22] | 2016 | Retrospective study | India | MICU and CCU | CPIS | 23.54 | NA |
| Mathai et al. [23] | 2016 | Prospective study | India | MICU and SICU | CDC | 40.1 | NA |
| Mathur et al. [24] | 2015 | Surveillance | India | NA | CDC | 17.07 | 34 |
| Mukhopadyay et al. [25] | 2010 | Prospective study | India | MICU | CPIS | 116 | 61.9 |
| Nakaviroj et al. [26] | 2014 | Retrospective study | Thailand | SICU | CDC | 8.21 | 33.33 |
| Parajuli et al. [27] | 2017 | Prospective study | Nepal | MICU and SICU | CDC | 21.4 | 34.7 |
| Park et al. [28] | 2014 | Retrospective study | South Korea | Cancer ICU | NA | 2.13 | NA |
| Rakshit et al. [29] | 2005 | Prospective cohort | India | CCU | CPIS | 26 | 37.5 |
| Ranjan et al. [30] | 2014 | Prospective study | India | MICU and CCU | CPIS | 31.7 | 48.3 |
| Recchaipichit kul et al. [31]* | 2013 | Surveillance study | Thailand | NA | CDC | 13.6/12.6 | NA |
| Rit et al. [32] | 2014 | Prospective study | India | NA | CPIS | 21.875 | NA |
| Sachdeva et al. [33] | 2017 | Prospective study | India | SICU | CDC | 25.11 | 74.7 |
| Singh et al. [34] | 2010 | Prospective study | India | MICU, SICU | CDC | 21.92 | NA |
| Singh et al. [35] | 2013 | Prospective stud | India | SICU | CDC | 32 | NA |
| Thongpiyapoom et al. [36] | 2004 | Prospective stud | Thailand | MICU and SICU | CDC | 10.8 | NA |

Note: SICU- Surgical Intensive Care Unit, MICU- Medical Intensive Care Unit, CCU- Critical Care Unit.

NA- Not available

* = Study with data of two consecutive years.

**Microbiology of VAP.** Twenty-four studies included data on microbiology, causing VAP, as shown in Table 2. Gram-negative organisms caused the majority of VAP episodes, followed by Gram-positive organisms. Only a few studies isolated other species like fungi causing VAP episodes.

Comparison among microbiology of VAP of 24 studies (Table 2) showed Acinetobacter sp., followed by Pseudomonas aeruginosa and then Klebsiella pneumoniae as common Gram-negative organisms causing VAP while Staphylococcus aureus and Enterococcus species as common Gram-positive organisms. Some studies also isolated sensitive and resistant forms of Gram-positive bacteria like MSSA (Methicillin Sensitive Staphylococcus Aureus), MRSA

**Table 2. The frequency of micro-organism isolated causing VAP episodes.**

| Study | Year | Gram Negative Organisms | | | | | | Gram Positive Organism | | | | Others |
| --- | --- | --- | --- | --- | --- | --- | --- | --- | --- | --- | --- | --- |
| | | Acinetobacter spp. | Klebsiella pneumoniae | Escherichia coli | Pseudo monas aeruginosa | Burkholderi a cepacia | Entero bacter sp. | Staphyloc occus aureus | MRSA | Enteroco ccus spp. | MSSA | Candid as pp. |
| Chittawatanarat et al. [13] | 2014 | 38.70% | 17.30% | 4% | 16.70% | - | 4.70% | 4% | - | 0.70% | - | 7.70% |
| Dasgupta et al. [14] | 2015 | - | 15.40% | 15.40% | 45.50% | - | - | - | - | 7.70% | - | - |
| | | | | | | | | | | | | |
| Datta et al. [15] | 2014 | 41.30% | 15.20% | - | 34.70% | - | - | - | - | - | - | - |
| Joseph et al. [16] | 2009 | 21.30% | - | - | 21.30% | - | - | 14.90% | - | 6.50% | - | - |
| Khan et al. [17] *** | 2016 | 18 | 12 | 4 | 18 | 2 | - | - | - | - | - | - |
| Khurana et al. [18] | 2017 | 54% | 13% | 3% | 21% | 1% | - | 3% | - | - | - | 0.70% |
| Kumar et al. [19] | 2017 | 40% | - | - | - | - | - | - | - | - | - | - |
| Mallic k et al. [20] | 2015 | 52% | 16% | 8% | 32% | - | - | - | 8% | - | - | - |
| Maqbool et al [21]* | 2017 | - | - | - | - | - | - | - | - | - | - | - |
| Masih et al. [22] | 2016 | 20 | 22% | - | 30% | - | - | - | 26% | 3% | - | 3% |
| Mathai et al. [23] | 2016 | 53.20% | 15.60% | 8.25% | 12.80% | - | - | - | 3.60% | - | 0.90% | - |
| Mathur et al. [24] | 2015 | 54% | 13% | 3% | 21% | 1% | - | 3% | - | - | - | 5% |
| Mukhoupadhya et al. [25] | 2010 | 76% | 14% | 14% | 43% | - | - | - | 9% | - | - | - |
| Nakaviroj et al. [26] | 2014 | 66.60% | - | 4.80% | 19.04% | - | - | - | 9.50% | - | - | - |
| Parajuli et. al. [27] | 2017 | 43% | 25% | 13.80% | 8.30% | 6.90% | - | - | - | - | - | - |
| Park et al. [28] | 2014 | 15.40% | - | - | 30.80% | - | - | - | 30.80% | - | - | - |
| Rakshit et al. [29] | 2005 | 8% | 29% | 13% | 46% | - | - | 25% | - | - | - | - |
| Ranjan et al. [30] | 2014 | 32.86% | 21.43% | 1.43% | 25.71% | - | 4.28% | 2.85% | - | 1.43% | - | - |
| Recchaipic hitkul et al. [31] ** | 2013 | 26.9%/38.9% | 15.4%/12.3% | 4.8%/2,9% | 25%/22% | - | 8.8%/6.3% | - | 7.2% /6% | - | 0.80% | - |
| Rit et al. [32] | 2014 | 17.60% | - | - | 30.70% | - | 35.20% | - | 7.60% | - | 11.70% | - |
| Sach deva et al. [33] | 2017 | 32% | 24% | - | 26.66% | - | - | 2 | - | 1 | - | - |
| Singh et al. [34] * | 2010 | - | - | - | - | - | - | - | - | - | - | - |
| Singh et al. [35] * | 2013 | - | - | 44444444 | - | - | 444444444- | - | - | - | - | - |
| Thon gpiya poom et al. [36] *** | 2004 | 5 | 2 | - | 4 | - | 1 | 1 | - | - | - | - |

Note: MRSA = Methicillin Resistant Staphylococcus Aureus.

MSSA = Methicillin Sensitive Staphylococcus Aureus.

* = Studies that have listed the micro-organisms but frequency is not given.

** = Study with data of two years 2008/2009.

*** = Studies in which frequency of micro-organism isolated is given in numbers.

(Methicillin Resistant Staphylococcus Aureus) and VRE (Vancomycin Resistant Enterococci). Candida albicans was the most common fungal isolate described in only seven studies.

## Discussion

Our review highlights the situation of VAP among the populations of countries of WHO SEAR. We found a wide range with much variability in the VAP rate, ranging from 2.13 to 166 per thousand ventilator days among these countries, showing most studies with alarming VAP situations. High mortality rates were reported in various studies and majority of the VAP episodes were caused by Gram-negative organisms, followed by Gram-positive organisms.

Differences in advancement and availability of health facilities, economic status, study setting (Medical or Surgical ICU) [37], criteria to diagnose VAP, patient characteristics diagnosed with various diseases, and medical staff practices in different regions of SEAR may be the most probable cause for variability in VAP rate.

A review by Arabi et al. among adults in developing countries showed incidence rates ranging from 10 to 41.7 per 1000 ventilator-days [38]. While Bonnel et al. estimated the VAP incidence in twenty-two Asian countries, including China, and showed a lower VAP incidence in high-income countries than lower-income countries (9 vs. 18.5 per 1,000 ventilator-days respectively) [39]. Studies from different parts of Asia (Qatar, Lebanon, Arabian Gulf countries, Iran, Japan) also showed a varied incidence rate from 4.8 to 12.6 per thousand [40–44]. Ding et al. also showed an incidence rate of 22.83 patients per 1000 ventilator days in China's mainland [45].

Some of the studies in our review have a similar incidence rate of VAP as of the above studies, but the remaining studies have a higher incidence. Various studies of Asia found out trauma, steroid use, enteral feeding, nasogastric tube placement, tracheostomy, reintubation, central venous catheter, blood transfusion, and COPD as risk factors for high VAP [46,47]. Similarly, globally found risk factors were aging males, increased ventilation time, consciousness disorders like swallowing, coughing; complications of burns, chronic disease, long time prophylactic use of antibiotics, and gene polymorphisms along with smoking [48]. Controlling these risk factors among patients can reduce incidence and mortality of VAP.

We found the highest incidence rate among ICU admitted patients of an Indian hospital [25], while the lowest was found in a South Korean hospital [28]. Possible reasons for the high rate may be due a smaller number of patients and study duration, and more male to female ratio. A review article found the male gender as a potential risk factor for VAP [48]. The study in a South Korean hospital equipped with advanced health care facilities showed the least VAP rate among cancer patients using antibiotics, antacid, along with health care staffs using a ventilator bundle approach [28].

We found a variable rate of VAP among different ICUs. Different infection control practices, injured and trauma patients with increased risk and medical staff's practices are probable reasons [49–51]. There is a lack of an acceptable gold standard for diagnosing VAP. Because of high sensitivity, the diagnostic modality of CPIS criteria is accepted widely. An application of this diagnostic modality is cost-effective and useful in a low resource setting like most SEAR countries [52,53].

Our study showed mortality rate ranged from 16.2% to 74.17%. The highest mortality rate was reported from among Indian patients [33]. We found that surgical ICU has more mortality than the medical ICU but cannot be generalized. This rate is similar to a study among developing countries by Arabi et al. In the USA mortality rate was reported by the Infectious Diseases Society of America (IDSA) and the American Thoracic Society (ATS) as 13%, and in Europe 30-day mortality rate was 29.9% [54,55] less than the countries of WHO SEAR. High

mortality in many areas in our study settings is due to lack of medical advancements in health care, lack of specialized ICU unit, and cost burden of personal and high antibiotic resistance [39].

Data on microbiology is similar to Arabi et al. showing Gram-negative bacilli as the most common pathogen (41–92%), followed by Gram-positive cocci (6–58%) [38]. An analysis done in Egypt also showed that similar common causative organisms; Pseudomonas aeruginosa, Klebsiella, Escherichia coli, Staphylococcus Aureus, Acinetobacter spp., Candida spp., and Proteus spp. were the most common microorganisms isolated while other organisms isolated were MRSA, Streptococci, Polymicrobial, Coagulase negative Staphylococci (CoNS), VRSA and MSSA [56].

As most WHO SEAR countries are low and middle income with a high VAP rate, a cost-effective educational intervention program is needed to reduce VAP. Use of heat and moisture exchanger (HME) vs. heated humidifying system (HHS), Staff education program, Hand hygiene training, and feedback program, awareness program and training on proper handling of respiratory secretions of critical ICU patients [57–59] are some of the cost-effective measures applied in these countries. Also, a ventricular bundle approach (head of bed elevation, peptic ulcer disease prophylaxis, deep venous thrombosis prophylaxis, and oral decontamination with chlorhexidine 0.12%) can be adopted for reducing VAP incidence [60]. Antibiotics are chosen by physicians, according to the knowledge of local microbiology causing VAP and their susceptibility patterns [61]. To keep in check the increasing drug resistant organisms with limited antibiotic inventory, infection control and antibiotic stewardship programs are mainstay strategies being adopted [62,63]. Thus, this review will motivate more surveillance and intervention studies to find risk factors and preventive strategies for VAP in countries of SEAR, as most countries do not have data on VAP. The outcomes chosen in our review provides a suitable overview to address the magnitude and scope of problem of VAP that helps us to know either our progress towards solving problems of VAP or the enough resources, manpower and economy to control and prevent infection or reduce the antibiotic resistance.

## Strengths and limitations

The major strength of our study is that this is the first systematic review conducted in this part of world exploring a major problem needed to be addressed. Our study has several shortcomings. Our study covered data of only five countries in this region which is the main limitation of our study. Another limitation was exclusion of non-English articles. While, we have also not described about outcomes such as early onset VAP and late onset VAP. Lastly, articles reporting incidence of ventilator associated Pneumonia parameters other than per thousand ventilator days were not included.

## Conclusion

Our review found a variable incidence of VAP in WHO SEAR regions, a comparatively alarming situation in most of these region's resource-limited countries with increasing mortality. As VAP is a critical issue in ICU with a high-cost burden with emerging antibiotic resistance, various interventional educational programs like staff training, hygiene awareness; use of ventilator bundle approach and surveillance programs along with addressing the possible risk factors warrants active participation from physicians, health workers to hospital administration and policymakers. Similarly, accurate and comprehensive testing of antimicrobial susceptibility and continuous monitoring along with implementation of antibiotic stewardship can possibly reduce the future risk of VAP in this region.

## Supporting information

**S1 Table. PRISMA checklist for systematic review and meta-analysis.**
(PDF)

**S1 File. Search strategy of PUBMED and EMBASE.**
(PDF)

## Acknowledgments

We want to acknowledge Dr. Ravi Pradhan and Dr. Siddhartha Bhandari for proofreading the manuscript.

## Author Contributions

**Conceptualization:** Sanjeev Kharel, Anil Bist.

**Methodology:** Sanjeev Kharel, Anil Bist.

**Resources:** Sanjeev Kharel.

**Supervision:** Shyam Kumar Mishra.

**Validation:** Shyam Kumar Mishra.

**Writing – original draft:** Sanjeev Kharel, Anil Bist.

**Writing – review & editing:** Sanjeev Kharel, Shyam Kumar Mishra.

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
