## [Decision Letter · Decision Letter 0]

1 Feb 2021

PONE-D-20-34750

Ventilator-Associated Pneumonia Among ICU patients in WHO Southeast Asian Region: A systematic review

PLOS ONE

Dear Dr.Sanjeev Kharel 

Thank you for submitting your manuscript to PLOS ONE. After careful consideration, we feel that it has merit but does not fully meet PLOS ONE’s publication criteria as it currently stands. Therefore, we invite you to submit a revised version of the manuscript that addresses the points raised during the review process.

We look forward to receiving your revised manuscript.

Kind regards,

Eleni Magira

Academic Editor

PLOS ONE

Journal Requirements:

2. We note that Figure 2 in your submission contains map images which may be copyrighted. All PLOS content is published under the Creative Commons Attribution License (CC BY 4.0), which means that the manuscript, images, and Supporting Information files will be freely available online, and any third party is permitted to access, download, copy, distribute, and use these materials in any way, even commercially, with proper attribution. For these reasons, we cannot publish previously copyrighted maps or satellite images created using proprietary data, such as Google software (Google Maps, Street View, and Earth). For more information, see our copyright guidelines: http://journals.plos.org/plosone/s/licenses-and-copyright.

(1) You may seek permission from the original copyright holder of Figure 2 to publish the content specifically under the CC BY 4.0 license. 

Reviewers' comments:

Reviewer's Responses to Questions

**Comments to the Author**

1. Is the manuscript technically sound, and do the data support the conclusions?

Reviewer #1: Partly

Reviewer #2: Partly

2. Has the statistical analysis been performed appropriately and rigorously? 

Reviewer #1: N/A

Reviewer #2: No

3. Have the authors made all data underlying the findings in their manuscript fully available?

Reviewer #1: Yes

Reviewer #2: Yes

4. Is the manuscript presented in an intelligible fashion and written in standard English?

Reviewer #1: Yes

Reviewer #2: Yes

Reviewer #1: In this systematic review of the literature on ventilator associated pneumonia in the South East Asian Region the Authors aimed to describe incidence, mortality and pathogens in order to implement strategies to prevent and control this disease. Twenty-four papers were selected following Prisma recommendations. VAP incidence range between 2.13 and 116 per thousand days. Mortality range from 16.2% and 74.1%. Acinetobacter spp, Pseudomonas aeruginosa, Klebsiella pneumoniae, Staphylococcus aureus and Enterococcus species are frequently involved. The Authors suggest different strategies to prevent VAP. This topic is relevant and interesting especially in this part of the world but several shortcomings need to be addressed:

Introduction:

- Include a specific reference line 41-42

Methods:

- Specify whether a registration has been made to PROSPERO or to another international registry for systematic reviews. Is essential

- Outcome are very essentials and maybe a proper research and description of risk factors should be included to increase the relevant of the study

Result

- Do not begin the sentence with an Arabic number line 128 and 130

- Results about possible risk factors included

Discussion

- The first paragraph of the discussion should sum the main results of the study: incidence, mortality etiological agents

- Please include something about the methods of the systematic review: for example, why these outcomes were chosen and not others?

- Clarify strengths and limitations of this research!

- Interventions to prevent and control VAP are speculative only and not systematically included in the results, however they are stated in the scope.

---

## [Author Response · Author response to Decision Letter 0]

11 Feb 2021

February 11, 2021.

Dr. Joerg Heber

Editor-in-Chief

PLOS ONE journal (open access)

We thank you for the time, effort and consideration you have put into our manuscript (PONE-D-20-34750) entitled " Ventilator-Associated Pneumonia Among ICU patients in WHO Southeast Asian Region: A systematic review ".

We would also like to express our utmost gratitude to the reviewer for his/her time and their valuable suggestions. Please find our response to the comments below:

Editors comments:

We thank the editor for the time and valuable feedback.

1.Comment: Please ensure that your manuscript meets PLOS ONE's style requirements, including those for file naming. The PLOS ONE style templates can be found at.

Response: We have checked our revised manuscript as per PLOS ONE’s style requirements, including those for file naming and edited if needed.

2.Comment: We note that Figure 2 in your submission contains map images which may be copyrighted. All PLOS content is published under the Creative Commons Attribution License (CC BY 4.0), which means that the manuscript, images, and Supporting Information files will be freely available online, and any third party is permitted to access, download, copy, distribute, and use these materials in any way, even commercially, with proper attribution. For these reasons, we cannot publish previously copyrighted maps or satellite images created using proprietary data, such as Google software (Google Maps, Street View, and Earth). For more information, see our copyright guidelines:

Response: Since, we are unable to obtain permission from the original copyright holder to publish these figures; have decided to remove the figure.

Reviewer

We thank the reviewer for the time and for the valuable feedback.

Reviewer response to answers:

1. Is the manuscript technically sound, and do the data support the conclusions?

Reviewer #1: Partly

Reviewer #2: Partly

Response: We have tried to address this comment by adding points in our revised manuscript in the conclusion section in Page no 33.

2.Has the statistical analysis been performed appropriately and rigorously?

Reviewer #1: N/A

Reviewer #2: No

Response: Our review article only qualitatively assessed the incidence, mortality and causative organisms of Ventilator Associated Pneumonia with no need of any statistical analysis.

3.Have the authors made all data underlying the findings in their manuscript fully available?

Reviewer #1: Yes

Reviewer #2: Yes

Response: Thank you for your positive response.

4. Is the manuscript presented in an intelligible fashion and written in standard English?

PLOS ONE does not copy edit accepted manuscripts, so the language in submitted articles must be clear, correct, and unambiguous. Any typographical or grammatical errors should be corrected at revision, so please note any specific errors here.

Reviewer #1: Yes

Reviewer #2: Yes

Response: Thank you for your positive feedback.

Reviewer 1 comments:

1.Comment: In this systematic review of the literature on ventilator associated pneumonia in the South East Asian Region the Authors aimed to describe incidence, mortality and pathogens in order to implement strategies to prevent and control this disease. Twenty-four papers were selected following Prisma recommendations. VAP incidence range between 2.13 and 116 per thousand days. Mortality range from 16.2% and 74.1%. Acinetobacter spp, Pseudomonas aeruginosa, Klebsiella pneumoniae, Staphylococcus aureus and Enterococcus species are frequently involved. The Authors suggest different strategies to prevent VAP. This topic is relevant and interesting especially in this part of the world but several shortcomings need to be addressed:

Response: Thank you for your encouraging comments.

2.Comment: Introduction: - Include a specific reference line 41-42

Response: A specific reference line in 41-42 is added as highlighted in the introduction section Page no 3 of revised manuscript.

3.Comment:Methods:

a. Specify whether a registration has been made to PROSPERO or to another international registry for systematic reviews. Is essential

 Response: We have mentioned our registration details under search strategy subsection of materials and methods section as highlighted in page no.6

b. Outcome are very essentials and maybe a proper research and description of risk factors should be included to increase the relevant of the study

Response: Yes indeed, outcomes related to description of risk factor is important but according to our inclusion criteria only studies reporting incidence, mortality and micro-organisms were included (page no. while most of the studies included in our review have no or limited information on risk factors. So, a brief description of risk factors was avoided but common risk factors from different studies conducted in Asian regions were assessed in discussion (Page no.29).

4.Comment: Result:

a. Do not begin the sentence with an Arabic number line 128 and 130

Response: The changes were made and highlighted in page no 8 and 9.

b. Results about possible risk factors included.

Response: Risk factors were not the outcomes chosen as per in our eligibility criteria. But further risk factors were reviewed briefly in discussion on page no.29(added information is highlighted). 

5.Comment: Discussion:

a. The first paragraph of the discussion should sum the main results of the study: incidence, mortality etiological agents

 Response: The first paragraph of discussion was changed as highlighted in page no.28

b. Please include something about the methods of the systematic review: for example, why these outcomes were chosen and not others?

Response: Thank you for your comment. We have included information on the methods used in our systematic review highlighted in the last paragraph of the discussion section in page no. 32.

c. Clarify strengths and limitations of this research!

Response: Thank you for your suggestions. The strengths and limitations of this research was addressed under strengths and limitations section above conclusion highlighted in page no. 32-33

d. Interventions to prevent and control VAP are speculative only and not systematically included in the results, however they are stated in the scope.

Response: Indeed, interventions to prevent and control VAP are speculative only but not included in results because of little or no information in our included articles. So, various intervention programs are mentioned as per the previous literature done in similar settings in the discussion section of page no. 31. 

We would once again like to thank the reviewers and editors for their generous and insightful comments to improve the paper.

Sincerely,

Mr. Sanjeev Kharel

Corresponding Author.

---

## [Decision Letter · Decision Letter 1]

15 Feb 2021

Ventilator-Associated Pneumonia Among ICU patients in WHO Southeast Asian Region: A systematic review

PONE-D-20-34750R1

Dear Dr. Sanjeev Kharel

We’re pleased to inform you that your manuscript has been judged scientifically suitable for publication and will be formally accepted for publication once it meets all outstanding technical requirements.

Kind regards,

Eleni Magira

Academic Editor

PLOS ONE

Additional Editor Comments (optional):

Reviewers' comments:

Reviewer's Responses to Questions

**Comments to the Author**

1. If the authors have adequately addressed your comments raised in a previous round of review and you feel that this manuscript is now acceptable for publication, you may indicate that here to bypass the “Comments to the Author” section, enter your conflict of interest statement in the “Confidential to Editor” section, and submit your "Accept" recommendation.

Reviewer #2: All comments have been addressed

2. Is the manuscript technically sound, and do the data support the conclusions?

Reviewer #2: Yes

6Review Comments to the Author

Reviewer #: Thanks for your corrections. All the points have been addressed and the text now is more complete.

Best regards

---

## [Editor Report · Acceptance letter]

17 Feb 2021

PONE-D-20-34750R1 

Ventilator-Associated Pneumonia Among ICU patients in WHO Southeast Asian Region: A systematic review 

Dear Dr. Kharel:

I'm pleased to inform you that your manuscript has been deemed suitable for publication in PLOS ONE. Congratulations! Your manuscript is now with our production department. 

Kind regards, 

on behalf of

Dr. Eleni Magira 

Academic Editor

PLOS ONE